# On Invertibility of an Interconnected System Composed of Two Dynamic Subsystems

**Mei Zhang [1], Boutaïeb Dahhou [2] and Ze-Tao Li [1,\*]**

[1] Guizhou Provincial Key Laboratory of Internet Plus Collaborative Intelligent Manufacturing Electrical Engineering School, Guizhou University, Guiyang 550025, China; mzhang3@gzu.edu.cn
[2] LAAS, UPS, University de Toulouse, F-31400 Toulouse, France; boutaib.dahhou@laas.fr
\* Correspondence: ztli@gzu.edu.cn

**Abstract:** In this paper, the invertibility of an interconnected system that consists of two dynamic subsystems was studied. It can be viewed as the distinguishability of the impacts of local input on the final global output, that is to say, whether the input at the local level can be recovered uniquely under a given output at the global level and initial state. The interconnected system constitutes two dynamic subsystems connected in a cascade manner. In order to guarantee the invertibility of the studied system, a necessary and sufficient condition was established. On the condition that both individual subsystems are invertible, the invertibility of the global system can be guaranteed. In order to recover the local input which generates a given global output, an algorithm was proposed for the studied interconnected system. Numerical examples were considered to confirm the effectiveness and robustness of the proposed algorithm.

**Keywords:** invertibility; interconnected system; input distinguishability; high-level subsystem; low-level subsystem





## 1. Introduction of Interconnected System

With the rapid developments of modern techniques, interconnections are becoming very common in modern control systems. The interconnected system here refers to the system composed of interacting subsystems either due to physical system structure or purposes of convenient analysis. The system or process itself may be the result of a series connection, parallel connection, and feedback interconnection of various subsystems [1]. For instance, a modern system is usually interfaced with multiple sensors, actuators, and process components. Therefore, a typical system can be viewed as composed of at least three interconnecting subsystems—actuator, sensor, and process subsystems. In order to ensure the normal operation of the whole system, the functions of these three parts must be normal. In addition, each subsystem can also be regarded as a series of dynamic subsystems, because each subsystem itself can be viewed as a dynamic system. In all cases, the global plant, as well as each subsystem, can be analyzed at different levels, up to the component level, to assess the reliability of the entire plant. It is becoming increasingly important to study the interconnected systems in analyzing dynamic systems, due to the fact that it allows to investigating less complicated components to study the properties of a complex system [2]. However, because of the requirement of a large amount of identification data or deep physical insight, the modeling and analysis of interconnected systems are challenging [3].

The concerning problems of interconnected systems are of great importance from a theoretical and practical viewpoint and have been studied extensively. The published research results involve the problems of stability, observability, controllability, and invertibility of interconnected systems; for example, in [4–8]. As shown in [8], the study provided a method to distinguish several dynamic subsystems, and it proved that there are usually delays during information transmission. This will lead to the instability and oscillation of

these systems. Therefore, many investigations are devoted to the stability of these systems. As illustrated in [4], the study derives a condition to ensure input to the stable state of an interconnected system, that is, to guarantee input to the stable state of both subsystems. In addition to stability, the controllability of interconnected systems was also addressed. Networked control systems are widely used in various fields of engineering; for example, in power generation and distribution systems [9], automotive control systems [10], cooperative control of unmanned vehicles [11], etc. The forms of interconnected systems also vary in the existing research. In most studies on tackling power series, the form of analytic functions interconnections was employed, e.g., [12]. In [13], an interconnected system constituted with a nonlinear followed by a linear time-invariant dynamical system was analyzed. The observability of an interconnected system composed of a partial differential equation (PDE) and an ordinary differential equation (ODE) was discussed in [14]. Two Fliess operators' compositions were found in [15]. In [16], the interconnections of bilinear subsystems were discussed.

When studying the characteristics of interconnected systems, the interconnected system composed of cascaded subsystems has received extensive attention. Generally speaking, due to the limitation of computational availability, system complexity, or communication bandwidth in the practical engineering world, it is very difficult to analyze cascaded interconnected systems with centralized structures [17]. Therefore, increasing attention has been paid in recent years to the research of distributed or decentralized methods. However, distributed analysis becomes more challenging due to the interaction between subsystems and the limited information available to each subsystem. Therefore, a problem worthy of attention is whether it can be proved that, under certain conditions, the influence of the lower subsystem can be distinguished on the higher subsystem, so as to avoid the complete measurement of the local subsystem. This can be regarded as a problem of the system's invertibility because one of the important objectives of invertibility analysis is to prove that the input or unknown input of the control system is distinguished.

In the past 50 years, due to its important theoretical and practical significance, the invertibility of the system has been widely studied. The research on the invertibility of nonlinear systems began in reference [18]. In this paper, Silverman's structural algorithm was extended to multiple-input multiple-output (MIMO) nonlinear systems. After that, the algorithm was modified in reference [19], which aimed at covering more kinds of systems. Similar literature related to the extension of the algorithm can be found in papers [20–25]. The invertibility of the system in the literature is related to the distinguishability of the system. The distinguishability of two variables refers to their ability to produce recognizable output for a given system. Some concepts of distinguishability or invertibility can be found in the literature, as in [18,22,26,27]. For example, the problem of the invertibility of switched linear systems was produced by Vu and Liberzon in [28], in which they discussed the ability to determine the active mode of the system from the input and output data. The idea was further extended to a nonlinear system in [27] and applied to fault diagnosis in [29].

In the above studies, the analysis of cascaded nonlinear systems received less attention. However, it is usually very important to describe the properties of composite systems, especially when the subsystems are nonlinear [30]. This paper considered a cascaded interconnected system consisting of two dynamic subsystems for physical or analysis purposes. The interconnection of two physical devices means that some variables associated with the first device are also variables or impact variables associated with the second device. Specifically, the problem is to give a sufficient and/or necessary condition, under which, given initial states, a local input can produce a distinguishable output of an interconnected system constituting two nonlinear subsystems. The essence of the problem is whether it can be proved that, under certain conditions, the input of the lower subsystem has a significant impact on the output of the higher subsystem. In order to solve this problem, the invertibility of cascaded interconnected systems is derived. The left invertibility of the interconnected system is capable of ensuring that the impacts of local variables on

the global level are distinguishable. The property of distinguishability of two inputs or parameters refers to their capacity to generate different output signals for a given input signal. The discussion of invertibility is of great significance in practical engineering. For example, for the problem of fault detection and isolation (FDI), a significant way is to treat the fault as an unknown input, and the motivation of invertibility is actually to detect and isolate the input, that is, to identify the possible location and time of the fault in the system; for example, in [29,31–33]. The contribution of this paper mainly lies in that it emphasizes the importance of the influences of local internal dynamics (actuator) on the global dynamics of the control system. Thus, it provides a basis for allowing the analysis of less complex subcomponents to study the characteristics of the interconnected systems.

The paper is organized as follows: Section 2 is devoted to the definition of invertibility of an interconnected dynamic system. In Section 3, conditions are given to validate involving definitions. Then, the procedure of computation of the inverse of the interconnected system is presented in Section 4. Numerical simulations are carried out to verify the effectiveness and robustness of the proposed method in Section 5. Finally, discussions and conclusions are made in Section 6.

## 2. Inversion of Nonlinear Interconnected System

### 2.1. Modeling of the Interconnected System

An interconnected system consisting of two dynamic subsystems was considered, as shown in Figure 1. The system decomposition could either be due to its physical structure or the purpose of analysis; for example, the local subsystem could represent field devices, like an actuator or a sensor. The high-level subsystem could then be the process dynamics, like a heat exchanger. In this way, it allowed the analysis of less complex subcomponents to study the characteristics of the interconnected system. Therefore, when analyzing the whole system, it was of great significance to treat the field equipment as an independent dynamic subsystem. The case of an inverted pendulum on a cart can be a typical example: if we view two subsystems physically as a cart and an inverted pendulum, then they constitute an interconnected system.

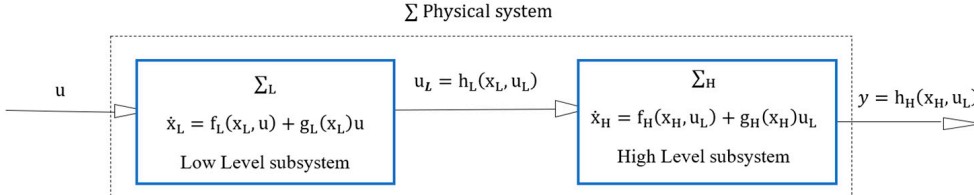

**Figure 1.** The structure of an interconnected system.

Assume that the low-level subsystem is an input affine dynamic system and depicted by (1):

$$\sum_L : \begin{cases} \dot{x}_L = f_L(x_L) + g_L(x_L)u, \ x_L(t_0) = x_{L0} \\ u_L = h_L(x_L) \end{cases} \tag{1}$$

where $x_L \in \mathcal{X}_L \subseteq \Re^L$ is the state, $u_L \in U_L \subseteq R^m$ is the output of the low-level subsystem, which is also the input of the high-level subsystem, and $u \in \mathcal{U} \subseteq \Re^l$ is the local input. $f_L$ and $g_L$ are smooth vector field on $\Re^n$ and $h$ is a smooth vector field on $\Re^m$. $f_L$, $g_L$, $h_L$ are algebraic functions, respectively.

An input affine dynamic system is also assumed for the higher level subsystem, and is described by (2):

$$\sum_H : \begin{cases} \dot{x}_H = f_H(x_H) + g_H(x_H)u_L, \ x_H(t_0) = x_{H0} \\ y = h_H(x_H) \end{cases} \tag{2}$$

where $x_H \in \mathcal{X}_H \subseteq \Re^n$ is the state of the high-level subsystem, $y \in \mathcal{Y} \subseteq \Re^p$ is the output of the high-level subsystem, which is also the output of the global system, and $u_L \in \mathcal{U}_L \subseteq$

$\Re^m$ is the input of the high-level subsystem, which is also the output of the low-level subsystem. $f_H$ and $g_H$ are smooth vector field on $\Re^n$ and $h_H$ is a smooth vector field on $\Re^p$. $f_H$, $g_H$, $h_H$ are algebraic functions, respectively.

In this way, combined with subsystems $\sum_L$ and $\sum_H$ in (1) and (2), an interconnected system $\sum$ was generated, where the input and output vectors were considered as $u$ and $y$, respectively.

For the interconnected system described by (1) and (2), the main objective of this paper is to give conditions to prove its invertibility. In this paper, invertibility refers to the ability to recognize and recover the input $u$ at the local level subsystem from the output $y$ at the global level. Therefore, the essence of invertibility in this paper is to investigate the capability of distinguishing local inputs from the influences on global final products. That is actually a study of the distinguishability of two inputs and their capabilities of producing different outputs.

### 2.2. Inverse of Interconnected System

In this section, the objective is to develop the required notations and provide some background knowledge related to the system invertibility. On this basis, the definition of correlated inverse of the interconnected system is given, and the formal problem statement is given. Then, the conditions to verify the relevant definitions are established.

#### 2.2.1. Inverse of Nonlinear System

This section describes the inverse of a dynamic system. The classical definition of invertibility for non-interconnected systems is given first. For the high-level subsystem depicted in (2), the input–output map is defined as $\mathcal{M}_H : \mathcal{U}_L \rightarrow \mathcal{Y}$, where the input function space is $\mathcal{U}_L$ and the corresponding output function space is $\mathcal{Y}$. $\mathcal{M}_H$ maps an input $u_L(.)$ to the output $y(.)$ generated by the system driven by $u_L(.)$ with an initial condition $x_{H0}$. The definition of invertibility of a nonlinear dynamical system is given below, as shown in [5].

**Definition 1.** *Fix an output set $\mathcal{Y}$ and consider an arbitrary interval $[t_0, T)$, the system (2) is invertible at a point $x_{H0} := x_H(t_0) \in \mathcal{X}_H$ over $\mathcal{Y}$, if for every $y_{[t_0, T)} \in \mathcal{Y}$, the equality $\mathcal{M}_{H(x_{H0})}\left(u_{L1[t_0, T)}\right) = \mathcal{M}_{H(x_0)}\left(u_{L2[t_0, T)}\right) = y$ implies that $\exists \varepsilon > 0$, such that $u_{L1[t_0, t_0+\varepsilon)} = u_{L2[t_0, t_0+\varepsilon)}$. The system is strongly invertible at a point $x_{H0}$ if it is invertible for each $x_H \in \mathcal{N}(x_{H0})$, where $\mathcal{N}$ is some open neighborhood of $x_{H0}$. The system is strongly invertible if there exists an open and dense sub-manifold $\mathcal{M}$ (called inverse sub-manifold) such that $\forall x_{H0} \in \mathcal{X}$, the system is strongly invertible at $x_{H0}$.*

In fact, by Definition 1, invertibility at $x_{H0}$ is equivalent to saying that $u_{L1[t_0, t_0+\varepsilon)} \neq u_{L2[t_0, t_0+\varepsilon)}$ for all $\varepsilon \in (0, T - t_0)$ implies that $\mathcal{M}_{H(x_0)}\left(u_{L1[t_0, T)}\right) = \mathcal{M}_{H(x_{H0})}\left(u_{L2[t_0, T)}\right)$. This notion was captured by Hirschorn in [18]. The concept of the inverse of nonlinear systems can now be extended to interconnected systems.

#### 2.2.2. Inverse of Nonlinear Interconnected System

As shown in Figure 2, the following problems are worth discussing: given the initial state and the corresponding output $y$ produced by the input $u$, under what conditions can the subsystems of the interconnected system uniquely recover the local input $u$? This problem is similar to the classical invertibility problem of a non-interconnected and nonlinear system.

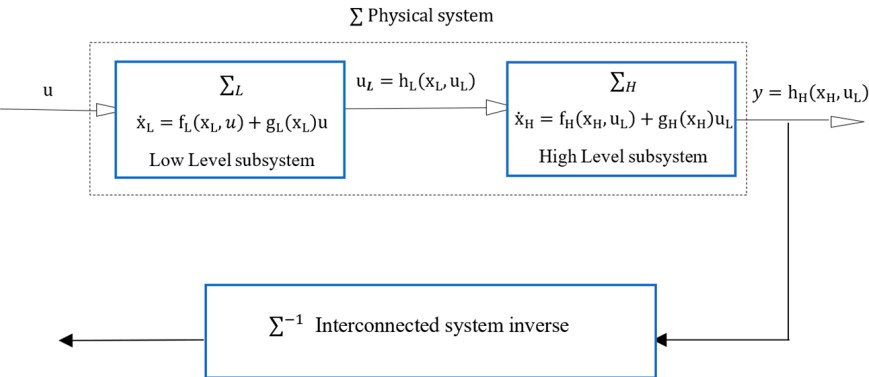

**Figure 2.** The inverse of an interconnected system.

Considering the input–output map of the high-level subsystem as $\mathcal{M}_H : \mathcal{U}_L \to \mathcal{Y}$, the input function space is generated by $\mathcal{U}_L$ and the corresponding output function space is generated by $\mathcal{Y}$, then $\mathcal{M}_H$ maps an input $u_L(.)$ to the output $y(.)$ generated by the system driven by $u_L(.)$ with an initial condition $x_{H0}$. In addition, the input–output map of the low-level subsystem is $\mathcal{M}_L : \mathcal{U} \to \mathcal{U}_L$ for some input function space $\mathcal{U}$ and the corresponding output $\mathcal{U}_L$. $\mathcal{M}_L$ maps an input $u(.)$ to the output $u_L(.)$ is generated by the system driven by $u(.)$ with an initial condition $x_{L0}$.

Define composition maps $\mathcal{M}_L \circ \mathcal{M}_H : \mathcal{U} \to \mathcal{Y}$ as the input–output map of the studied interconnected system, given initial conditions $(x_{L0}, x_{H0})$, the composition $\mathcal{M}_L \circ \mathcal{M}_H$ maps an input $u(.)$ to the output $y(.)$ driven by $u(.)$. The interconnected system is said to be invertible if driven input $u(t)$ can be recovered by the output $y(.)$, part of the state and the initial state $(x_{L0}, x_{H0})$. It is to extend the definition of invertibility to the cascade interconnected system by Definition 2 as follows:

**Definition 2.** *Fix an output set $\mathcal{Y}$ and consider an arbitrary interval $[t_0, T)$, the interconnected system described in (1) and (2) is invertible at a point $(x_{L0}, x_{H0}) = x_H(t_0) \in \mathcal{X}$ over $\mathcal{Y}, x_L(t_0) \in \mathcal{X}_L(t_0)$ over $\mathcal{U}_L$, if for every $y_{[t_0, T)} \in \mathcal{Y}$, the equality $(\mathcal{M}_L \circ \mathcal{M}_H)_{(x_{L0}, x_{H0})}\left(u_{1[t_0, T)}\right) = (\mathcal{M}_L \circ \mathcal{M}_H)_{(x_{L0}, x_{H0})}\left(u_{2[t_0, T)}\right) = y_{[t_0, T)}$ implies that $\exists \varepsilon > 0$, such that $u_{1[t_0, t_0+\varepsilon]} = u_{2[t_0, t_0+\varepsilon]}$. The system is strongly invertible at a point $(x_{L0}, x_{H0})$ if it is invertible for each $x_L \in \mathcal{N}_L(x_{L0})$, $x_H \in \mathcal{N}_H(x_{H0})$, where $(\mathcal{N}_L, \mathcal{N}_H)$ is some open neighborhood of $(x_{L0}, x_{H0})$. The system is strongly invertible if there exists an open and dense sub-manifold $\mathcal{M}_L$ of $\mathcal{X}_L$, $\mathcal{M}_H$ of $\mathcal{X}_H$, such that $\forall (x_{L0}, x_{H0}) \in (\mathcal{M}_L, \mathcal{M}_H)$, the system is strongly invertible at $(x_{L0}, x_{H0})$.*

The invertibility proposed in Definition 2 may not hold in two cases: on one hand, if there are two different inputs $u_1$, $u_2$ produce two equal outputs $u_{L1} = u_{L2}$, while for the high-level subsystem, $u_{L1}, u_{L2}$ are its inputs, clearly, these equal inputs will definitely yield equal outputs $y_1 = y_2$ at the global terminal; in this case, invertibility of the interconnected system fails to hold; on the other hand, even if the two different inputs $u_1$, $u_2$ generate two different local outputs $u_{L1}, u_{L2}$ at the low-level subsystem, these two $u_{L1}, u_{L2}$ yield equal output $y_1 = y_2$ at the global level, and again, the invertibility of the interconnected system cannot be guaranteed.

The former implies that the low-level subsystem is not invertible, while the latter may be caused by the non-invertibility of the high-level subsystem.

## 3. On the Condition of Invertibility of the Interconnected System

The system under consideration is an interconnected system, resulting in the unavailability of classical inversion technologies. It is therefore necessary to investigate a new toolset for guaranteeing the invertibility of interconnected systems. The problem of the inverse of interconnected systems can be regarded as a combination of invertible mappings and individual input recovery. Therefore, the basic idea of solving the invertibility

problem is to utilize the relationship between the output and the state of the subsystem to synthesize the mapping and then use the nonlinear structure algorithm to recover the input of the corresponding subsystem. It can be concluded from Definition 2 that the interconnected system cannot be invertible, no matter whether the high-level or low-level subsystem is invertible or not. Therefore, the necessary and sufficient conditions for invertibility of interconnected systems on set $\mathcal{U}$, $\mathcal{U}_L$, and $\mathcal{Y}$ can be given from the view of the individual subsystem.

**Theorem 1.** *Consider the interconnected system $\sum$ which consists of two subsystems: low-level $\sum_L$ and high-level $\sum_H$ subsystems depicted by (1) and (2), and an output set $\mathcal{Y}$. The interconnected system is invertible at $(x_{H0}, x_{L0})$ over $\mathcal{Y}$, if and only if each subsystem, the low-level $\sum_L$ and the high-level $\sum_H$, is invertible at $x_{L0}$ over $\mathcal{U}_L$, and $x_{H0}$ over $\mathcal{Y}$, respectively.*

**Proof.** Consider $\mathcal{M}_L$ as the input–output mapping of low-level $\sum_L$ subsystem, while $\mathcal{M}_H$ is the input–output mapping of high-level $\sum_H$ subsystem. Then the composition $\mathcal{M}_L \circ \mathcal{M}_H$ can be considered as the input–output mapping of the interconnected system.

In order to confirm the theorem, it needs to provide a condition from both sufficiency and necessity aspects. To begin with the sufficiency condition, the invertibility of a dynamic system refers to the bijective of the input–output mapping. Since both subsystems are invertible, the corresponding mapping $\mathcal{M}_L$ and mapping $\mathcal{M}_H$ are bijective mappings. Moreover, the composition of two bijective mappings is a bijective mapping, so input–output mapping $\mathcal{M}_L \circ \mathcal{M}_H$ of the cascade system is bijective. Thus, the cascade interconnected system is invertible.

The next task is to produce a necessary condition. It can be achieved by the non-invertibility of either subsystem. That is to say if any of the subsystems is not invertible at $(x_{L0}, x_{H0})$, then the interconnected system $\sum$ is also not invertible.

On one hand, suppose that the high-level subsystem $\sum_H$ is not invertible, then no matter whether the low-level subsystem $\sum_L$ is invertible or not, for the low-level subsystem described in (1), fix an output set $\mathcal{U}_L$ and consider an arbitrary interval $[t_0, T)$. Two distinct inputs exist for $\exists\, \varepsilon > 0\ u_1 \neq u_2$ on $[t_0, t_0 + \varepsilon)$, and it is possible to generate two different outputs $\mathcal{M}_{L(x_{L0})}\left(u_{1[t_0, T)}\right) = u_{L1[t_0, T)}$, $\mathcal{M}_{L(x_{L0})}\left(u_{2[t_0, T)}\right) = u_{L2[t_0, T)}$, $u_{L1[t_0, T)} \neq u_{L2[t_0, T)}$. However, for the high-level subsystem in (2), even if the invertibility of this subsystem is guaranteed, fix an output set $\mathcal{Y}$. These distinguishable inputs $u_{L1} \neq u_{L2}$ on $[t_0, t_0 + \varepsilon)$ may generate two equal outputs $\mathcal{M}_{H(x_{H0})}\left(u_{L1[t_0, T)}\right) = \mathcal{M}_{H(x_{H0})}\left(u_{L2[t_0, T)}\right) = y_{1[t_0, T)} = y_{2[t_0, T)} = y_{[t_0, T)}$. Thus, from the aspect of the global level, these two distinct original local inputs $u_1 \neq u_2$ on $[t_0, t_0 + \varepsilon)$ produce two equal global outputs $y_1 = y_2$ on $[t_0, t_0 + \varepsilon)$:

$$(\mathcal{M}_L \circ \mathcal{M}_H)_{(x_{L0},\ x_{H0})}\left(u_{1[t_0, T)}\right) = (\mathcal{M}_L \circ \mathcal{M}_H)_{(x_{L0},\ x_{H0})}\left(u_{2[t_0, T)}\right) = y_{[t_0, T)} \quad (3)$$

As a result, the interconnected system $\sum$ can not be invertible at $(x_{H0}, x_{L0})$ over $(\mathcal{U}, \mathcal{U}_L, \mathcal{Y})$.

For the other, suppose that the low-level subsystem $\sum_L$ is not invertible, then no matter whether the high-level subsystem $\sum_H$ is invertible or not, for the low-level subsystem $\sum_L$ in (1), fix an output set $\mathcal{U}_L$ and consider an arbitrary interval $[t_0, T)$. Two different local inputs exist for $\exists\, \varepsilon > 0\ u_1 \neq u_2$ on $[t_0, t_0 + \varepsilon)$ that can produce two equal outputs: $\mathcal{M}_{L(x_{L0})}\left(u_{1[t_0, T)}\right) = u_{L1[t_0, T)}$, $\mathcal{M}_{L(x_{L0})}\left(u_{2[t_0, T)}\right) = u_{L2[t_0, T)}$, $u_{L1[t_0, T)} = u_{L2[t_0, T)}$. From the aspect of global level, even if invertibility of the high-level subsystem $\sum_H$ in (2) is ensured, these equal inputs $u_{L1} = u_{L2}$ on $[t_0, t_0 + \varepsilon)$ are only capable of producing two equal outputs $\mathcal{M}_{H(x_{H0})}\left(u_{L1[t_0, T)}\right) = \mathcal{M}_{H(x_{H0})}\left(u_{L2[t_0, T)}\right) = y_{1[t_0, T)} = y_{2[t_0, T)} = y_{[t_0, T)}$ at the final level. Thus, for the cascade interconnected system, two different local inputs $u_1 \neq u_2$ on $[t_0, t_0 + \varepsilon)$ produce two equal global outputs $y_1 = y_2$ on $[t_0, t_0 + \varepsilon)$:

$$(\mathcal{M}_L \circ \mathcal{M}_H)_{(x_{L0},\ x_{H0})}\left(u_{1[t_0, T)}\right) = (\mathcal{M}_L \circ \mathcal{M}_H)_{(x_{L0},\ x_{H0})}\left(u_{2[t_0, T)}\right) = y_{[t_0, T)} \quad (4)$$

As a result, the interconnected system $\sum$ can not be invertible at $(x_{H0}, x_{L0})$ over $(\mathcal{U}, \mathcal{U}_L, \mathcal{Y})$. $\square$

**Theorem 2.** *Consider the interconnected system $\sum$ consists of two subsystems: the low-level $\sum_L$ and the high-level $\sum_H$ subsystems depicted by (1) and (2), and an output set $\mathcal{Y}$. The interconnected system is strongly invertible at $(x_{H0}, x_{L0})$ over $\mathcal{Y}$ if and only if each low-level $\sum_L$ and high-level $\sum_H$ subsytems is strongly invertible at $x_{L0}$ over $\mathcal{U}_L$, and $x_{H0}$ over $\mathcal{Y}$, respectively.*

**Remark 1.** *If the interconnected system $\sum$ depicted by (1) and (2) is globally invertible, then the local inputs can be uniquely recovered over the time interval $[t_0, T)$. Moreover, T can be arbitrarily large if the state trajectories do not show a finite escape time.*

From Theorem 2, in order to obtain the invertibility of the studied interconnected system, the key criterion is to ensure the invertibility of the individual subsystem. In a differential-algebraic setting, the left invertibility can be determined in terms of the differential output rank of the system (see [30,31,34])

**Definition 3.** *The differential output rank $\rho$ of a system is equal to the differential transcendence degree of the differential extension $k\langle y \rangle$ over the differential field $k$, i.e., $\rho = diff \, trd^{\circ} \, k\langle y \rangle \, /k$.*

**Property 1.** *The differential output rank $\rho$ of a system is smaller or equal to $\min(m, p)$*

$$\rho = \text{diff tr d}^{\circ} \text{k}\langle \text{y} \rangle \, /\text{k} \leq \min(\text{m, p}) \tag{5}$$

*where $m$, $p$ are the total number of inputs and outputs, respectively.*

The differential output rank $\rho$ is also the maximum number of outputs that are related by a differential polynomial equation with coefficients over $\mathcal{K}$ (independent of $x$ and $u$).

**Theorem 3.** *A system is left-invertible if and only if the differential output rank $\rho$ is equal to the total number of inputs, e.g., $\rho = m$ in (2).*

That is, if the differential output rank is equal to the number of the inputs, the system is invertible. This implies that the number of outputs must be greater, or equal to the number of inputs.

## 4. On Computation of the Dynamics Inverse of the Interconnected System

After verifying the invertibility of the interconnected system, it is capable of recovering the original inputs uniquely from the global measurement. It implies that each original local input affects the global output distinguishably. In fact, if a system is invertible, there are already structure algorithms that allow one to express the input as a function of the output, its derivatives, and possibly some states (for example, in [4,21,23,33]).

A methodology given in Theorem 1 is now capable of checking the invertibility of a nonlinear system. Considering the interconnected input–output system $\sum$ with two subsystems $\sum_L$ and $\sum_H$ from inputs $\mathcal{U}$ into outputs $\mathcal{Y}$, its composition input–output map is $\mathcal{M}_L \circ \mathcal{M}_H$. If the interconnected system is left invertible, there exists an input–output system $\sum^{-1}$ from inputs $\mathcal{Y}$ into outputs $\mathcal{U}$, and the inverse composition map is defined as $(\mathcal{M}_L \circ \mathcal{M}_H)^{-1}$, such that the cascade system $\sum \sum^{-1} : \mathcal{U} \to \mathcal{Y} \to \mathcal{U}$ is the identity. In our mainly algebraic setting, it was supposed that $\mathcal{U}, \mathcal{Y}$ are a good class of functions equipped with an algebraic structure; for example, they are differential vector space. Then the inverse of the interconnected system is defined as in Theorem 3.

**Theorem 4.** *Consider the interconnected system $\sum$ that consists of two subsystems: the low-level $\sum_L$ and high-level $\sum_H$ subsystems, and an input–output set $(\mathcal{U}, \mathcal{Y})$. If the interconnected system*

*is strongly invertible at* $(x_{H0}, x_{L0})$ *over* $(\mathcal{U}_L, \mathcal{Y})$, *then the inverse interconnected system can also be an interconnected system with the input–output set* $(\mathcal{Y}, \mathcal{U})$, *as follows:*

$$(\mathcal{M}_L \circ \mathcal{M}_H)^{-1} = \mathcal{M}_H{}^{-1} \circ \mathcal{M}_L{}^{-1} \tag{6}$$

**Proof.** Supposed that both $\mathcal{M}_L$ and $\mathcal{M}_H$ are invertible, then $\mathcal{M}_L{}^{-1} \circ \mathcal{M}_H = i_{\mathcal{U}}$, $\mathcal{M}_H{}^{-1} \circ \mathcal{M}_H = \mathcal{M}_L \circ \mathcal{M}_L{}^{-1} = i_{\mathcal{U}}$ and $\mathcal{M}_H \circ \mathcal{M}_H{}^{-1} = i_{\mathcal{Y}}$. $\square$

Thus, for any $y \in \mathcal{Y}$, one can obtain

$$\begin{aligned} \left[(\mathcal{M}_L \circ \mathcal{M}_H) \circ \left(\mathcal{M}_H{}^{-1} \circ \mathcal{M}_L{}^{-1}\right)\right](y) &= \left[\mathcal{M}_L \circ \left(\mathcal{M}_H \circ \mathcal{M}_H{}^{-1}\right) \circ \mathcal{M}_L{}^{-1}\right](u) \\ &= \mathcal{M}_L \circ \left(i_U\left(\mathcal{M}_L{}^{-1}(u)\right)\right) \\ &= \mathcal{M}_L \circ \mathcal{M}_L{}^{-1}(u) \\ &= i_U(u) \end{aligned}$$

It follows that $(\mathcal{M}_H \circ \mathcal{M}_L) \circ \left(\mathcal{M}_L{}^{-1} \circ \mathcal{M}_H{}^{-1}\right) = i_{\mathcal{Y}}$. Similarly, one can show that $\left(\mathcal{M}_L{}^{-1} \circ \mathcal{M}_H{}^{-1}\right) \circ (\mathcal{M}_H \circ \mathcal{M}_L) = i_{\mathcal{U}}$. Therefore, $\mathcal{M}_L \circ \mathcal{M}_H$ is invertible with inverse $\mathcal{M}_H{}^{-1} \circ \mathcal{M}_L{}^{-1}$.

It can be seen from Figure 3 that the inverse system of an interconnected system can also be regarded as an interconnected system, and its components are the inverse subsystems of each subsystem. In the interconnected inverse system, the global output is the original input of the interconnected system. This implies that it is capable of distinguishing the impacts of each local input on the global output. To compute the inverse of an input affine interconnected system, the structure algorithm allows us to express the input as a function of the output, its derivatives, and possibly some states, as shown in [4].

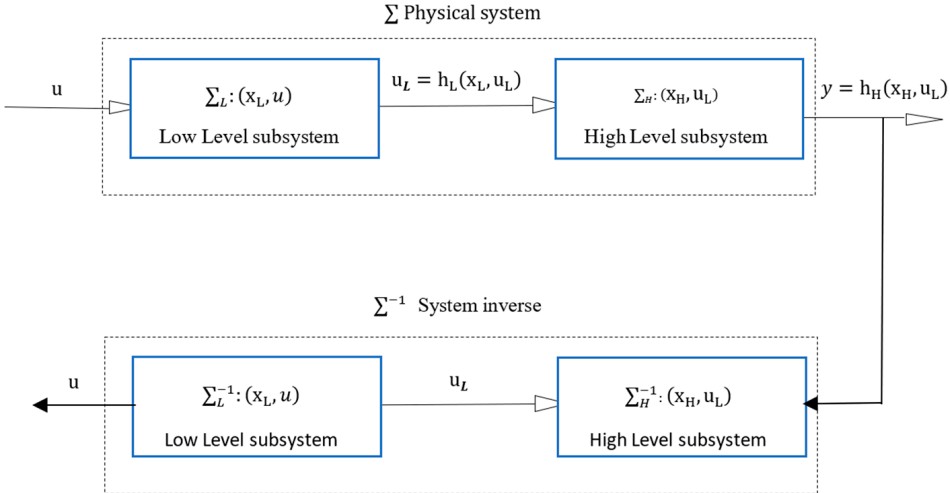

**Figure 3.** Scheme of the inverse of an interconnected system.

For the high-level subsystem depicted in (2), the expression of its inverse dynamics can be realized as from (7):

$$\mathcal{M}_H{}^{-1} : \begin{cases} \dot{\eta}_H = \varphi_H(\eta_H, y, \dot{y}, \ldots) \\ u_L = \omega_H(\eta_H, y, \dot{y}, \ldots) \end{cases} \tag{7}$$

where $\eta_H$ is a function of the state $x_H$ of the high-level subsystem.

For the low-level subsystem depicted in (1), the expression of its inverse dynamics can be realized as from (8):

$$\mathcal{M}_L^{-1} : \begin{cases} \dot{\eta}_L = \varphi_L\left(\eta_L, u_L, \dot{u}_L, \ldots\right) \\ u = \omega_L\left(\eta_L, u_L, \dot{u}_L, \ldots\right) \end{cases} \tag{8}$$

where $\eta_L$ is a function of the state $x_L$ of the low-level subsystem.

Inverse dynamics (7), together with inverse dynamics (8), constitutes the inverse of the studied interconnected system. It can be seen that the basis of the proposed calculation of the inverse of the interconnected dynamic system is the existence of left invertibility of the original interconnected cascade system. The feasibility of the input-based inverse reconstruction method is determined by the existence of left invertibility. For the interconnected inverse system, the output is the input of the original system, while the input is the output of the original global system and its possible time derivative. A series of invertibility related analysis has shown that a key point is the concept of relative degree. The theory is suitable for linear time-invariant and nonlinear systems with vector relativity. More details about the relative degree can be found in reference [4].

**Definition 4.** *(Relative degree of nonlinear systems). For the invertible dynamic system described by (2), the relative degree $r_i$ of the output $y_i$ with respect to the input vector $u_L$ is the smallest integer which is defined by*

$$L_{g_{Hj}} L_{f_H}^{r_i - 1} h_{Hi}(x_H) \neq 0; \ 1 \leq j \leq m \tag{9}$$

$$L_{g_{Hj}} L_{f_H}^{k} h_{Hi}(x_H) = 0; \ 0 \leq k < r_i - 1, \ 1 \leq j \leq m \tag{10}$$

*where $L_{f_H}(.)$ and $L_{g_H}(.)$ represent the Lie derivatives of a real function $h_H(x_H)$ along the vector field $f_H(x_H)$ and $g_H(x_H)$.*

$$L_{f_H}^{0} h_{Hi}(x_H) = h_{Hi}(x_H)$$
$$L_{f_H}^{k} h_{Hi}(x_H) = \frac{\partial\left(L_{f_H}^{k-1} h_{Hi}(x_H)\right)}{\partial x_H} f_H(x_H)$$
$$\text{and } L_{g_{Hj}} L_{f_H}^{k} h_{Hi}(x_H) = \frac{\partial\left(L_{f_H}^{k} h_{Hi}(x_H)\right)}{\partial x_H} g_{Hj}(x_H).$$

Denote a matrix $\Delta(x_H)$ as

$$\Delta(x_H) = \begin{bmatrix} L_{g_{H1}} L_{f_H}^{r_1 - 1} h_{H1}(x_H) & \cdots & L_{g_{Hm}} L_{f_H}^{r_1 - 1} h_{H1}(x_H) \\ \cdots & \cdots & \cdots \\ L_{g_{H1}} L_{f_H}^{r_m - 1} h_{Hm}(x_H) & \cdots & L_{g_{Hm}} L_{f_H}^{r_m - 1} h_{Hm}(x_H) \end{bmatrix} \tag{11}$$

$\Delta(x_H)$ is a nonsingular matrix with full rank:

$$\text{rank } \Delta(x_H) = m \tag{12}$$

In order to derive a function of states and output in (2) to represent $u_L(t)$ as, the first step is to differentiate $y_i$ $i = 1, \ldots, m$ to obtain the derivatives:

Suppose $r_i = 1$, one can get:

$$y_i^{(1)} = \frac{\partial h_{Hi}(x_H)}{\partial x_H} \dot{x}_H(t)$$
$$= \frac{\partial h_{Hi}(x_H)}{\partial x_H}\left(f_H(x_H) + g_H(x_H) u_L\right)$$
$$= L_{f_H}^{1} h_{Hi}(x_H) + \sum_{j=1}^{m} L_{g_{Hj}}^{1} L_{f_H}^{0} h_{Hi}(x_H) u_{Lj}$$

If $r_i \neq 1$, then $L^1_{g_{Hj}} L^0_{f_H} h_{Hi}(x_H) = 0; \ 1 \leq j \leq m$

In this way,

$$y_i^{(1)} = L^1_{f_H} h_{Hi}(x_H) \tag{13}$$

Generally speaking, it needs to continue this differential procedure, for $k < r_i$, one will have

$$y_i^{(j)} = L^j_{f_H} h_{Hi}(x_H)$$
$$= \partial_{x_H}\left(L^{j-1}_{f_H} h_{hi}(x_H) f_H(x_H)\right) + \sum_{s=0}^{j-2} \partial_{u_L^{(j)}}(L^{j-1}_{f_H} h_{Hi}(x_H)) u_L^{(s)} \ j = 0,\ldots,k, \ k < r_i \tag{14}$$

Until the relative degree $r_i$, it reaches

$$y_i^{(r_i)} = L^{r_i}_{f_H} h_{Hi}(x_H) + \sum_{j=1}^{m} L_{g_{Hj}}(L^{r_i-1}_{f_H} h_{hi}(x_H)) \ u_{Lj} \ i = 1,\ldots,m \tag{15}$$

If there are *m* relative order $r_1,\ldots,r_m$ related to the output $y$, and the total relative degree satisfied (16):

$$r = \sum_{i=1}^{m} r_i = n \tag{16}$$

then calculating expressions for their derivatives can be referred to as a one-step algorithm to obtain an inverse, and we get

$$\begin{bmatrix} y_1^{(r_1)} \\ \vdots \\ y_m^{(r_m)} \end{bmatrix} = \begin{bmatrix} L^{r_1}_{f_H} h_{H1}(x_H) \\ \vdots \\ L^{r_m}_{f_H} h_{Hm}(x_H) \end{bmatrix} + \begin{bmatrix} L_{g_{H1}} L^{r_1-1}_{f_H} h_{H1}(x_H) & \cdots & L_{g_{Hm}} L^{r_1-1}_{f_H} h_{H1}(x_H) \\ \cdots & \cdots & \cdots \\ L_{g_{H1}} L^{r_m-1}_{f_H} h_{Hm}(x_H) & \cdots & L_{g_{Hm}} L^{r_m-1}_{f_H} h_{Hm}(x_H) \end{bmatrix} u_L \tag{17}$$

the Equation (18) can be solved for $u_L$ to obtain

$$u_L = \begin{bmatrix} L_{g_{H1}} L^{r_1-1}_{f_H} h_{H1}(x_H) & \cdots & L_{g_{Hm}} L^{r_1-1}_{f_H} h_{H1}(x_H) \\ \cdots & \cdots & \cdots \\ L_{g_{H1}} L^{r_m-1}_{f_H} h_{Hm}(x_H) & \cdots & L_{g_{Hm}} L^{r_m-1}_{f_H} h_{Hm}(x_H) \end{bmatrix}^{-1} \cdot \left( \begin{bmatrix} y_1^{(r_1)} \\ \vdots \\ y_m^{(r_m)} \end{bmatrix} - \begin{bmatrix} L^{r_1}_{f_H} h_{H1}(x_H) \\ \vdots \\ L^{r_m}_{f_H} h_{Hm}(x_H) \end{bmatrix} \right) \tag{18}$$

In this situation, there will be no internal dynamics, and all the results will be finite time in nature, see reference [34].

However, normally, the total relative degree is assumed:

$$r = \sum_{i=1}^{m} r_i < n \tag{19}$$

In this case, the system given by (2) can be presented on a new basis that is introduced as follows:

Define the following change of the coordinates:

$$\xi_{Hi} = \left[ \xi^1_{Hi}, \ \xi^2_{Hi}, \ \ldots, \xi^{r_i}_{Hi} \right]^T$$
$$= \left[ \phi^1_{Hi}(x_H), \ \phi^2_{Hi}(x_H), \ \ldots, \phi^{r_i}_{Hi}(x_H) \right]^T$$
$$= \left[ h_{Hi}(x_H), L_{f_H} h_{Hi}(x_H), \ldots, L^{r_i-1}_{f_H} h_{Hi}(x_H) \right]^T \ i = 1,\ldots,m$$
$$\xi_H = [\xi_{H1}, \xi_{H2}, \ldots, \ \xi_{Hm}]$$
$$= [\phi_{H1}(x_H), \ \phi_{H2}(x_H), \ \ldots, \ \phi_{Hm}(x_H)]$$
$$\zeta_H = \left[ \phi_{H(r+1)}(x_H), \ \phi_{H(r+2)}(x_H), \ \ldots, \ \phi_{Hn}(x_H) \right]^T$$
$$y = \left[ \xi^1_{H1}, \xi^1_{H2}, \ldots, \ \xi^1_{Hm} \right]$$

By applying the new local coordinates transformation proposed in [4], if the system holds the assumption of relative degree, it is always possible to find the function $\phi_{H(r+1)}(x_H)$, $\phi_{H(r+2)}(x_H)$, ..., $\phi_{Hn}(x_H)$, thus,

$$\phi_H(x_H) = \left[ \phi_{H1}(x_H),\ \phi_{H2}(x_H),\ \ldots,\ \phi_{Hm}(x_H), \phi_{H(r+1)}(x_H),\ \ldots,\ \phi_{Hn}(x_H) \right] \tag{20}$$

The mapping $\phi_H(x_H)$ is a local diffeomorphism, which means

$$x_H = \phi_H^{-1}(\xi_H, \zeta_H) \tag{21}$$

Furthermore, according to [5], if the assumption is satisfied,

**Assumption 1.** *The distribution* $\Gamma = span\left\{ \begin{array}{cccc} g_{H1} & g_{H2} & \cdots & g_{Hm} \end{array} \right\}$ *is involutive, then, it is always possible to identify the function* $\phi_{H(r+1)}(x_H)$, $\phi_{H(r+2)}(x_H)$, ..., $\phi_{Hn}(x_H)$ *in such a way that*

$$L_{g_{Hj}}\phi_{Hi}(x_H) = 0,\ i = r+1,\ldots,n,\ j = 1,\ldots,m$$

$$\dot{\zeta}_H = q(\xi_H, \zeta_H)$$

Then, the input vector $u_L$ can be obtained by means of the output vector $y$ and its derivatives:

$$u_L = \Delta\left(\phi_H^{-1}(\xi_H, \zeta_H)\right)^{-1} \left( \begin{bmatrix} \xi_{H1}^{(r_1)} \\ \vdots \\ \xi_{Hm}^{(r_m)} \end{bmatrix} - \begin{bmatrix} L_{f_H}^{r_1} h_{H1}\left(\phi_H^{-1}(\xi_H, \zeta_H)\right) \\ \vdots \\ L_{f_H}^{r_m} h_{Hm}\left(\phi_H^{-1}(\xi_H, \zeta_H)\right) \end{bmatrix} \right) \tag{22}$$

Fortunately, along with the discussion of this paper, linear and nonlinear problems can be treated in parallel with each other. Results for linear time-invariant (LTI) systems will always be viewed as special cases of the results obtained for the nonlinear problems specified by the general input affine nonlinear system model.

## 5. Numerical Simulations

In this section, numerical simulation was employed to validate the effectiveness and robustness of the proposed algorithm. The main objective is to confirm, by means of numerical simulations, that the input of an invertible interconnected system can be recovered uniquely from the measured global output. A case study was developed on an intensified heat exchanger (HEX). More relative information can be found in [35]. During the course of the simulation work, the aim is to prove that the pneumatic pressure of the actuators at the local level can be recovered by the measured outlet temperatures of HEX at the global level.

### 5.1. Modeling of the Interconnected System

5.1.1. Low-Level Subsystem Modeling

The pneumatic control valve was employed to act as an actuator in this system. In [36,37], a pneumatic control valve can be modeled in the following form:

$$p_c A_L = m\frac{d^2v}{dt} + \mu\frac{dv}{dt} + kv \tag{23}$$

where $A_L$ is the diaphragm area on which the pneumatic pressure acts, $p_c$ is the pneumatic pressure, $m$ is the mass of the control valve stem, $\mu$ is the friction of the valve stem, $k$ is the spring compliance, and $v$ is the stem displacement or percentage opening of the valve.

$$x_L{}^T = \begin{bmatrix} x_{L1} & x_{L2} & x_{L3} & x_{L4} \end{bmatrix} = \begin{bmatrix} v_1 & \frac{dv_1}{dt} & v_2 & \frac{dv_2}{dt} \end{bmatrix},$$

$$u^T = \begin{bmatrix} u_1 & u_2 \end{bmatrix} = \begin{bmatrix} P_{c1} & P_{c2} \end{bmatrix}, \quad u_L{}^T = \begin{bmatrix} Q_1 & Q_2 \end{bmatrix} = \begin{bmatrix} g_v\sqrt{\frac{P_1}{sg}}v_1 & C_v\sqrt{\frac{\Delta P_2}{sg}}v_2 \end{bmatrix},$$

$$C = \begin{bmatrix} c_1 & c_2 & c_3 & c_4 \end{bmatrix} = \begin{bmatrix} g_v\sqrt{\frac{P_1}{sg}} & 0 & g_v\sqrt{\frac{P_2}{sg}} & 0 \end{bmatrix}$$

the actuator subsystem is then described by four states, two inputs and two outputs, as

$$\begin{cases} \dot{x}_L = \begin{bmatrix} 0 & 1 & 0 & 0 \\ -\frac{k_1}{m} & -\frac{\mu_1}{m} & 0 & 0 \\ 0 & 0 & 0 & 1 \\ 0 & 0 & -\frac{k_2}{m} & -\frac{\mu_2}{m} \end{bmatrix} x_L + \begin{bmatrix} \frac{A_L}{m} & 0 \\ 0 & 0 \\ 0 & \frac{A_L}{m} \\ 0 & 0 \end{bmatrix} u \\ u_L = \begin{bmatrix} g_v\sqrt{\frac{P_1}{sg}} & 0 & g_v\sqrt{\frac{P_2}{sg}} & 0 \end{bmatrix} x_L \end{cases} \tag{24}$$

### 5.1.2. High-Level Subsystem Modeling

The HEX can be modeled based on the mass and energy balances that describe the evolution of the characteristic values—temperature, mass, composition, pressure, etc. From [16], the dynamic equations governing the heat balance of the process fluid and the utility fluid are given by

$$\begin{cases} \dot{T}_H = \frac{UA}{\rho_H c_{pH} V_H}(T_C - T_H) + \frac{1}{V_H}\left(T_H^{in} - T_H\right)Q_H \\ \dot{T}_C = \frac{UA}{\rho_C c_{pC} V_C}(T_H - T_C) + \frac{1}{V_C}\left(T_C^{in} - T_C\right)Q_C \end{cases} \tag{25}$$

where $\rho_H$, $\rho_C$ are the density of the process fluid and utility fluid (kg.m$^{-3}$), $V_H$, $V_C$ are volume of the process fluid and utility fluid (m$^3$), $C_{p_H}, C_{p_C}$ are specific heat of the process fluid and utility fluid (J.kg$^{-1}$.K$^{-1}$), $U$ is the overall heat transfer coefficient (J.m$^{-2}$.K$^{-1}$.s$^{-1}$), $A$ is the reaction area (m$^2$), $Q_H$, $Q_C$ are mass flowrate of the process fluid and utility fluid (kg.s$^{-1}$), $T_H$ is the process fluid temperature, $T_H^{in}$ is the inlet temperature of the process fluid, $T_C$ is the utility fluid temperature, and $T_C^{in}$ is the inlet temperature of utility fluid.

Define the state vector as $x_H{}^T = \begin{bmatrix} x_{H_1}, x_{H2} \end{bmatrix}^T = [T_H, T_C]^T$, the control input $u_L{}^T = [u_{L1}, u_{L2}]^T = [Q_H, Q_L]^T$, the output vector of measurable variables $y_H{}^T = [y_{H1}, y_{H2}]^T = [T_H, T_C]^T$, then the above two equations can be rewritten in the following state-space form:

$$\begin{cases} \dot{x}_H = f_H(x_H) + \sum_{i=1}^{2} g_{Hi}(x_H)u_L \\ y_H = h_H(x_H, u_L) \end{cases} \tag{26}$$

where $f_H(x_H) = \begin{pmatrix} f_{H1}(x_H) \\ f_{H2}(x_H) \end{pmatrix} = \begin{pmatrix} \frac{h_H A}{\rho_H C_{p_H} V_H}(T_H - T_C) \\ \frac{h_u A}{\rho_C C_{p_C} V_C}(T_C - T_H) \end{pmatrix}$, and $g_H = (g_{H1}, g_{H2}) =$

$\begin{pmatrix} \frac{(T_H^{in} - T_H)}{V_H} & 0 \\ 0 & \frac{(T_C^{in} - T_C)}{V_C} \end{pmatrix}$, $y_{H1} = x_{H1}$, $y_{H2} = x_{H2}$, $T_H^{in}$, $T_C^{in}$ are measured and constant.

An interconnected system is constituted by (26) and (27).

### 5.2. Invertibility Checking

As mentioned above, one of the key points in computing dynamic inverses through system inverses is the reversibility of systems. The above discussed algebraic criteria were employed. Then, the system inputs can be expressed as a function of the output and its derivative. In order to test the invertibility of the interconnected systems modeled by (25) and (26), the output differential rank of each subsystem is required to equal the number of the inputs.

First, the invertibility of the high-level subsystem should be checked. As shown in (25), there are two inputs, $u_{L1}, u_{L2}$, representing the fluid flow rates $Q_H$, $Q_C$ of both fluids, respectively. In order to verify the invertibility characteristics, the output differential rank should be equal to the input number 2. To achieve this purpose, the explicit expressions of input and output $y_H$ are derived by calculating the derivative $y_H$. Corresponding to (26), there are two outputs, $y_{H1}, y_{H2}$, representing the temperature of both fluids $T_H$ and $T_C$, differentiating all two outputs, the following equation is obtained:

$$\begin{cases} \dot{y}_{H1} = \frac{h_H A}{\rho_H C_{P_H} V_H}(y_{H2} - y_{H1}) + \frac{u_{L1}}{V_H}\left(T_H^{in} - y_{H1}\right) \\ \dot{y}_{H2} = \frac{h_u A}{\rho_C C_{P_C} V_C}(y_{H1} - y_{H2}) + \frac{u_{L2}}{V_C}\left(T_C^{in} - y_{H2}\right) \end{cases} \tag{27}$$

From Equation (27), all the output differential equations are dependent on states $x_H$ and inputs $u_L$, thus there are $r = 0$ independent relations, outputs number $p$ is 2, then the output differential rank is as follows:

$$\rho = p - r = 2 \tag{28}$$

According to Theorem 3, the invertibility of the high-level subsystem can be verified.

After that, the invertibility of the low-level subsystem should be checked. There are two local inputs $u_1$, $u_2$, in (25), representing the pneumatic pressure $p_{c1}$, $p_{c2}$ of both fluids. By differentiating the two outputs $u_{L1}$, $u_{L2}$ and finding all possible independent relations, it is easy to check that the low-level subsystem is invertible.

*5.3. Inverse System Representing*

In order to reconstruct inputs of both high-level and low-level subsystems, they are represented as a function of the global measurement outputs and their derivatives. According to the structure algorithm introduced in Section 4, an expression for the two inputs of the high-level subsystem can be derived as $\widetilde{u}_L = \begin{bmatrix} \widetilde{u}_{L1} & \widetilde{u}_{L2} \end{bmatrix}$.

$$\begin{cases} \widetilde{u}_{L1} = \frac{V_p}{T_H^{in} - y_{H1}}\left(\dot{y}_{H1} - \frac{h_H A}{\rho_H C_{P_H} V_H}y_{H2} + \frac{h_H A}{\rho_H C_{P_H} V_H}y_{H1}\right) \\ \widetilde{u}_{L2} = \frac{V_u}{T_C^{in} - y_{H2}}\left(\dot{y}_{H2} - \frac{h_u A}{\rho_C C_{P_C} V_C}y_{H1} + \frac{h_u A}{\rho_C C_{P_C} V_C}y_{H2}\right) \end{cases} \tag{29}$$

Then an expression for the two original inputs of the low-level subsystem can be derived as $u = \begin{bmatrix} u_1 & u_2 \end{bmatrix}$.

$$\begin{cases} u_1 = \alpha.\beta_1\left[\ddot{\widetilde{u}}_{L1} + \gamma_{11}\dot{\widetilde{u}}_{L1} + \gamma_{12}\widetilde{u}_{L1}\right] \\ u_2 = \alpha.\beta_2\left[\ddot{\widetilde{u}}_{L2} + \gamma_{21}\dot{\widetilde{u}}_{L2} + \gamma_{22}\widetilde{u}_{L2}\right] \end{cases} \tag{30}$$

where $\alpha = \frac{m}{A_a}$, $\beta_i = 1/g_v\sqrt{\frac{P_i}{sg}}$, $i = 1, 2, \gamma_{11} = -\frac{k_1}{m}, \gamma_{12} = -\frac{\mu_1}{m}, \gamma_{21} = -\frac{k_2}{m}, \gamma_{22} = -\frac{\mu_2}{m}$.

*5.4. Numerical Simulations and Discussion*

The proposed computation algorithm of invertibility of the interconnected system was confirmed by simulations with the values from [16]. These relevant values were as follows: Inlet temperatures of both fluids $T_H^{in}$ and $T_C^{in}$ were 78 °C and 18 °C. Both fluid flow rates were the interconnection of the global system and were assumed to be not measured, their expected values were obtained by theoretical computation, and the specific computed values were $4.22 \times 10^{-5}$ m³ s⁻¹ for $Q_C$ of the utility fluid and $4.17 \times 10^{-6}$ m³ s⁻¹ for $Q_H$ of the process fluid, respectively. Parameters related to fluids actuators were $m = 2$ kg, $A_a = 0.029$ m², $\mu = 1500$ Ns/m and $k = 6089$ Ns/m, pressure drop $\Delta P$ in utility fluid was 0.6 MPa and 60 KPa in the process fluid. The pneumatic pressures $p_{c1}$, $p_{c2}$ were considered as the two inputs of the low-level subsystem, the values were 1 MPa and 1.2 Mpa. The purpose of the design is to testify whether the value of the pneumatic pressure recovered by the inverse of the interconnected system is consistent with the original value. In order to achieve this purpose, two simulations were carried out. In case 1, fixed values

of both inputs were considered, by contrast in case 2, an abrupt change was supposed to apply to the input $p_{c2}$, the pneumatic pressure of the utility fluid. Simulation results are reported in Figures 4–15. In order to prove the robustness, external interference or measurement noise was considered at the sensor of temperature of process fluid $T_H$ in the simulation. The colored noise was generated with a second-order AR filter excited by Gaussian white noise with zero mean and unitary variance.

### 5.4.1. Case 1: Both Pneumatic Pressure $p_{c1}$, $p_{c2}$ Are Fixed

Both local inputs of pneumatic pressures $p_{c1}$, $p_{c2}$ were considered to be fixed in this case. The simulation purpose is to confirm if the local inputs pneumatic pressure $p_{c1}$, $p_{c2}$ can be uniquely recovered by the global measured temperatures $T_H$, $T_C$, under given initial conditions. Figures 4–9 confirm the reconstructability and robustness of the interconnected system.

Figures 4 and 5 show the measured temperatures of both fluids under noise-free and noise-corrupted situations. From Figures 4a and 5a, it can be seen that, after a relatively short transient time, the temperatures of both fluids stabilized at a new level, and from Figures 4b and 5b, it can be seen that even measurements were corrupted by the colored noise, convergence can also be ensured. These measurements were then employed to reconstruct inputs of high-level subsystems.

As shown in Figure 6, the expected computed value of process fluid flow rates $Q_H$ are plotted in the black solid lines, and red dash lines represent the reconstructed values via the inverse of the high-level subsystem. It is illustrated in Figure 6a that, after a short transient period, if the measurement is not corrupted by noise, the reconstructed value in the red dash curve tracks the computed value in the black solid curve correctly. And from Figure 6b, when measurement suffers noise, the reconstructed value can also track the computed one with acceptable accuracy. It is obvious that the inverse of the high-level subsystem is capable of recovering its inputs with acceptable accuracy.

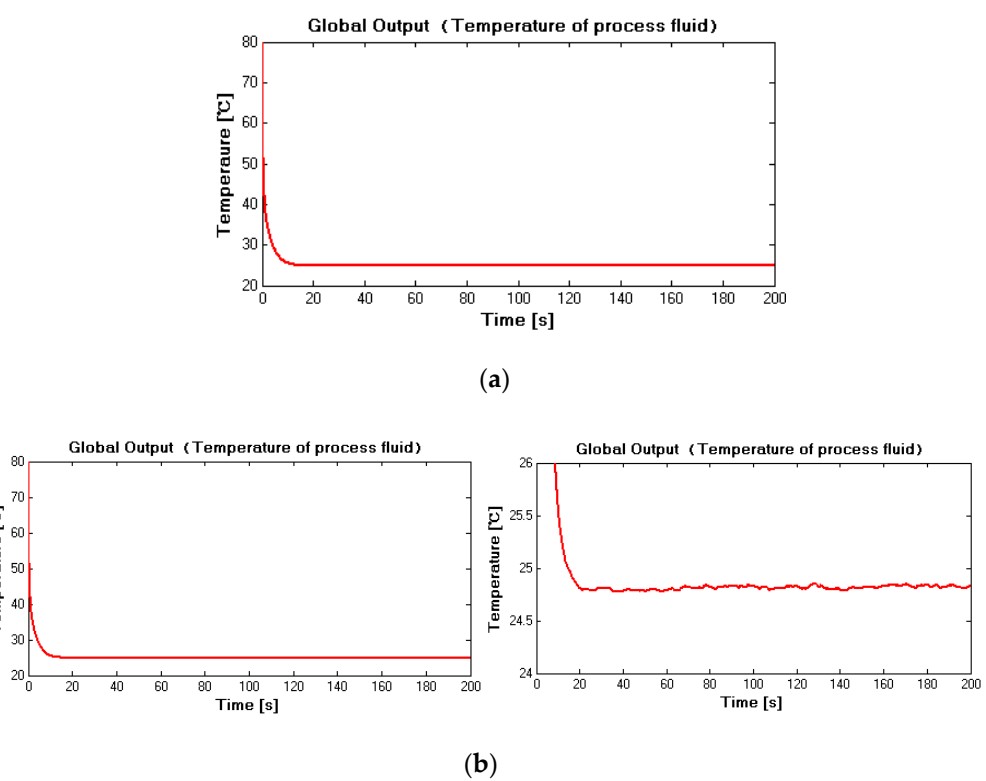

**Figure 4.** (**a**) Measured $T_H$ (noise-free situation) in case 1; (**b**) measured $T_H$ (noise-corrupted situation) in case 1.

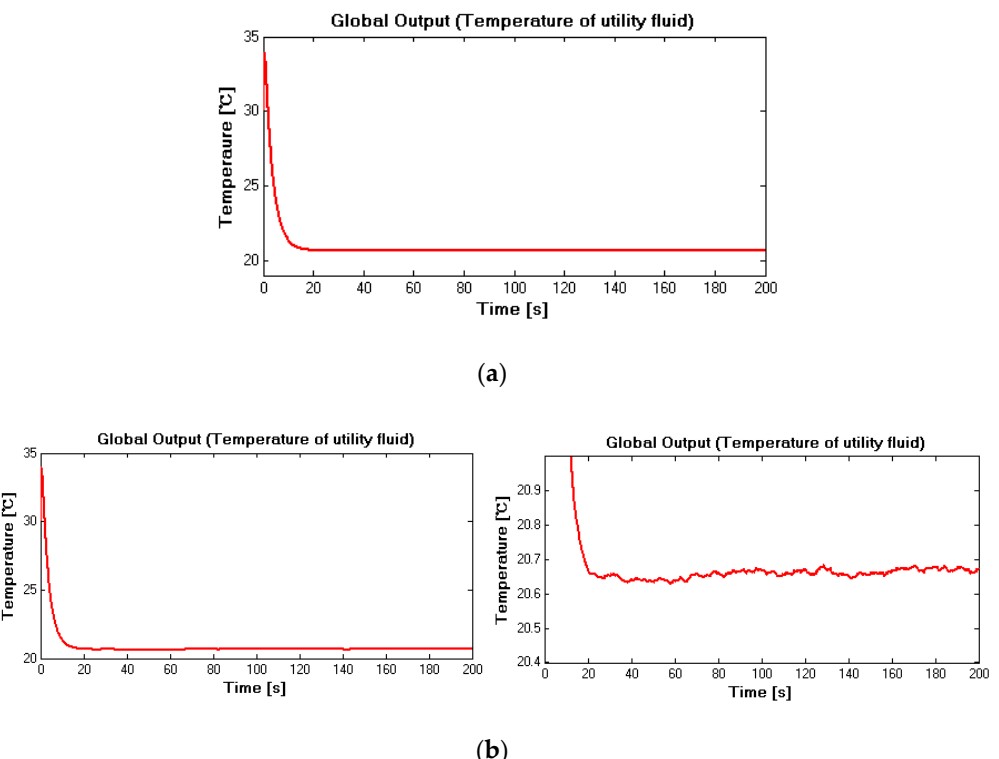

**Figure 5.** (**a**) Measured $T_C$ (noise-free situation) in case 1; (**b**) measured $T_C$ (noise-corrupted situation) in case 1.

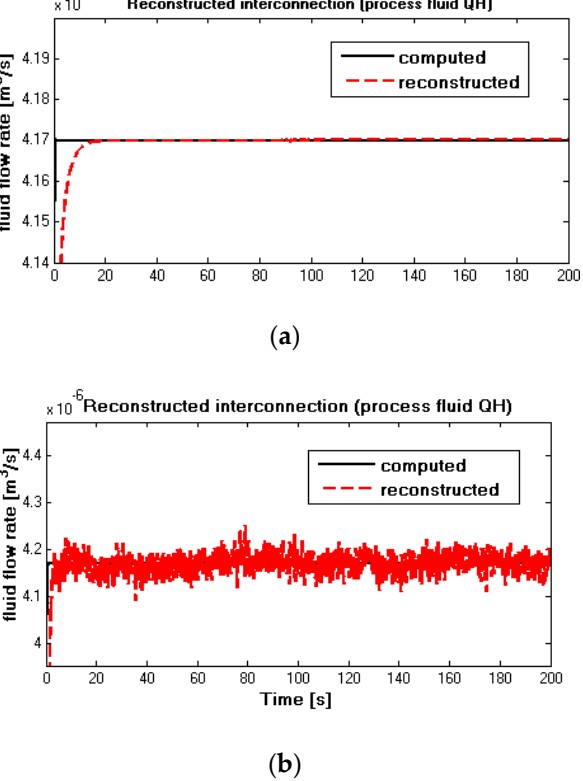

**Figure 6.** (**a**) Reconstructed and computed $Q_H$ (noise-free situation) in case 1; (**b**) reconstructed and computed $Q_H$ (noise-corrupted situation) in case 1.

When it comes to the utility fluid case in Figure 7a,b, the computed value in black solid lines overlapped the reconstructed values in red dash lines after a short transient time, no matter with or without measurement noise. Figure 7a illustrates the noise-free case while Figure 7b represents the noise-corrupted situation. From Figure 7b, compared with measured temperature, it can be seen that noise value impacts reconstruct value more significantly. Whatever the case, the input reconfiguration capability of the high-level subsystem is confirmed. Thereafter, these reconstructed values were used as inputs of the inverse of the low-level subsystem, with the aim of checking the reconfiguration capacities.

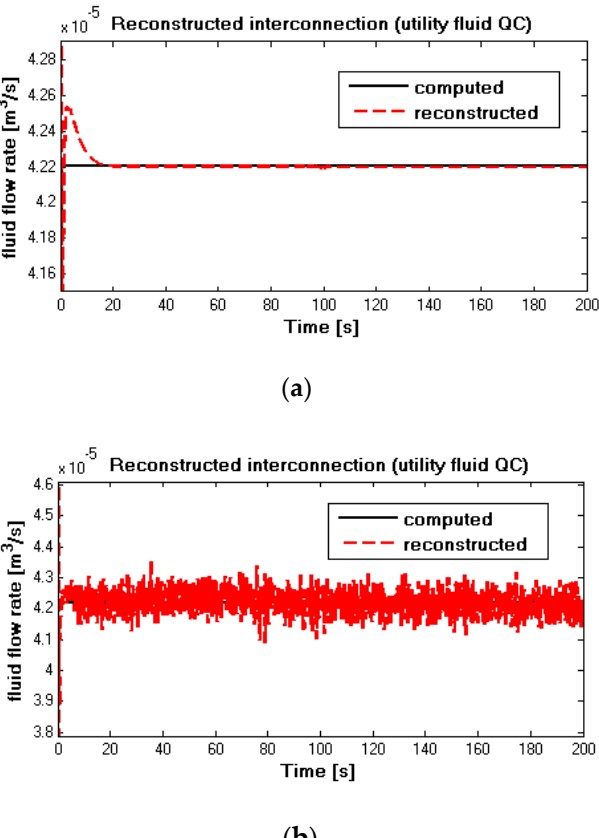

**Figure 7.** (**a**) Reconstructed and computed $Q_C$ (noise-free situation) in case 1; (**b**) reconstructed and computed $Q_C$ (noise-corrupted situation) in case 1.

It can be seen from Figure 8a,b that the input reconfiguration capability of the low-level subsystem is confirmed. It is obvious in Figure 8a,b that the recovered pneumatic pressure of the process fluid $p_{c1}$ in red dash lines converge to the measurement value in black solid lines correctly, and the short transient time is less than 2 s. From Figure 8a, during the first two seconds, oscillation is observed. It can be seen from Figure 8b that the reconstructed value is influenced clearly by measurement noise. Since recovered pneumatic pressure $p_{c1}$ is also the original input of the interconnected system, thus reconfiguration capability of the global interconnected system is also confirmed.

For utility fluid flowrate, similar results to process fluid are obtained. From Figure 9a,b, it can be seen that the recovered pneumatic pressure $p_{c2}$ in red dash lines converge to the measured pneumatic pressure in black solid curves after a short transient time, which are also the outputs of the inverse interconnected system.

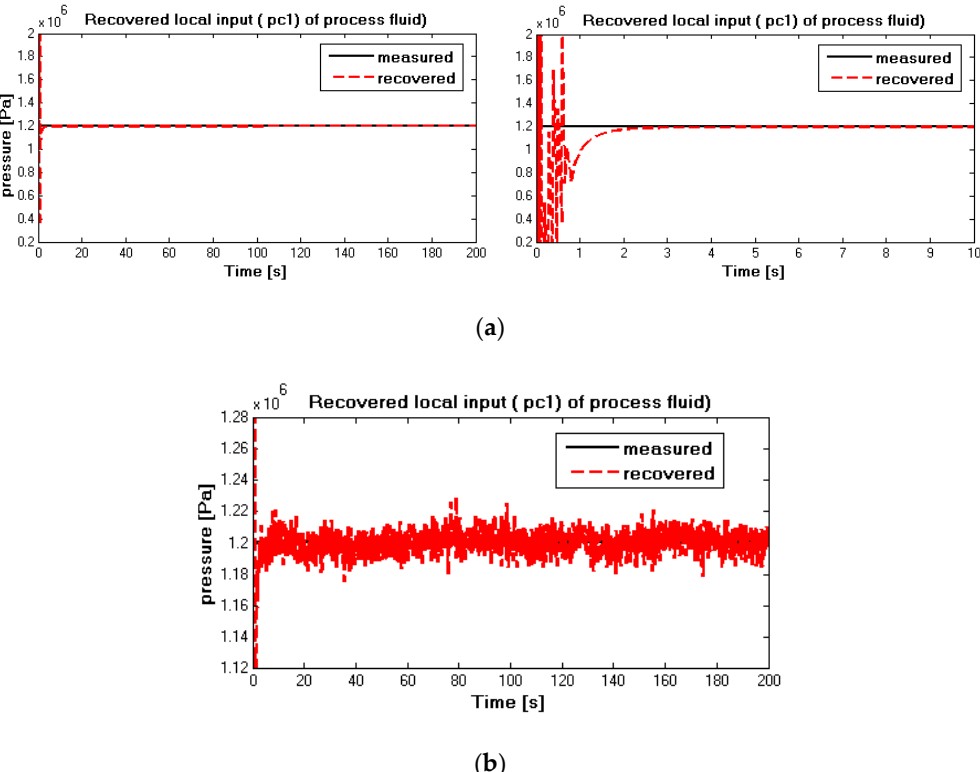

**Figure 8.** (**a**) Measured and recovered $p_{c1}$ (noise-free situation) in case 1; (**b**) measured and recovered $p_{c1}$ (noise-corrupted situation) in case 1.

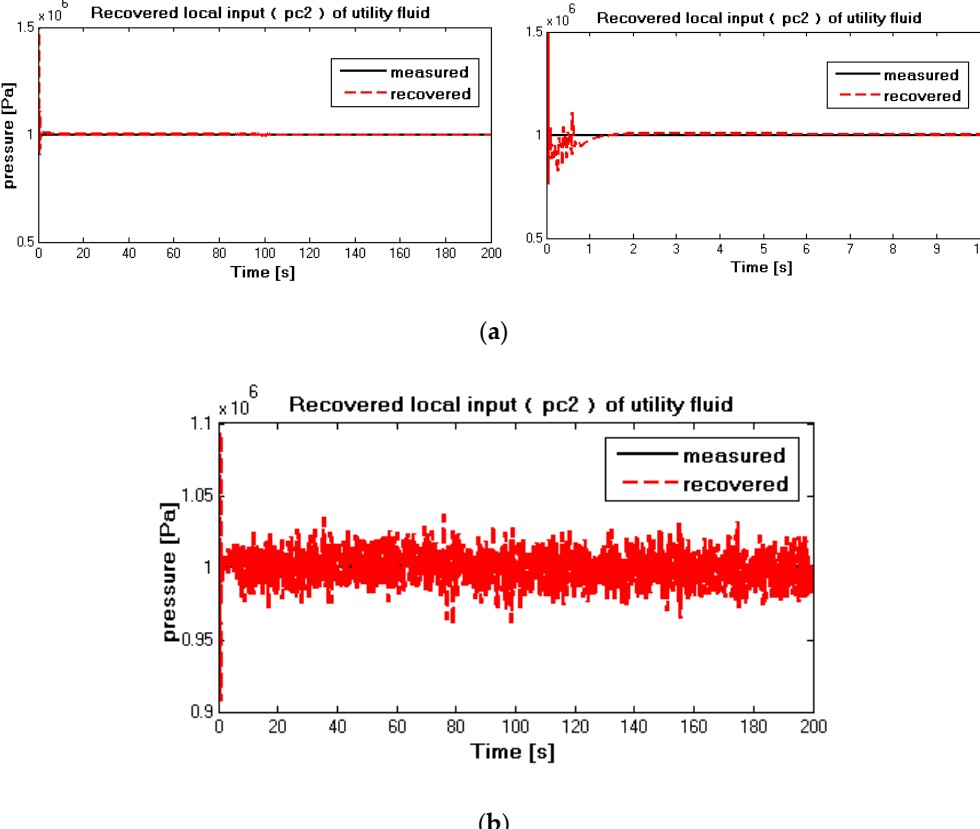

**Figure 9.** (**a**) Measured and recovered $p_{c2}$ (noise-free situation) in case 1; (**b**) measured and recovered $p_{c2}$ (noise-corrupted situation) in case 1.

In sum, it can be concluded that, if local inputs remain constant and both subsystems are invertible, the local inputs of an interconnected system can be recovered by the global system correctly, and both effectiveness and robustness are guaranteed.

5.4.2. Case 2: Pneumatic Pressure $p_{c1}$ Remains Fixed, Pneumatic Pressure $p_{c2}$ Increases Abruptly

In this case, the aim is to illustrate that the proposed local input recovered algorithm is also available even if the local inputs are time-varying. To achieve this purpose, a sudden increase was supposed to apply to the pneumatic pressure $p_{c2}$ of the utility fluid, where another 0.5 MPA was assumed at time 120 s. Figures 10–15 report the simulation results.

Figures 10 and 11 represent the measured temperature of both fluids, Figure 10a describes the noise-free situation, and Figure 10a,b means that measurement of process temperature $T_H$ is corrupted by the colored noise. It can be seen from Figures 10 and 11 that the temperature of both fluids at the global level varies at 120 s due to variation of pneumatic pressure $p_{c1}$ at the local level. The temperature of the utility fluid in Figure 10b is not influenced by noise since measured noise was supposed on the process fluid. Thus, from Figure 11b, it is clear that the measured process fluid temperature is fluctuant with a small amplitude due to noise. These measured fluid temperatures were fed to the high-level inverse subsystem to reconstruct inputs of the high-level subsystem.

From Figure 12a,b, it can be obtained that the reconstructed flow rates $Q_H$ of the process fluid in red dash curves overlap the computed values in black solid lines after a short transient time. Resulting from the supposed variation of pneumatic pressure $p_{c2}$ at 120 s, the reconstructed values fluctuate a little. For the noise-free case in Figure 12a, after less than two seconds, the reconstructed flow rate $Q_H$ of the process fluid converges back to the computed values in black solid lines again. For the noise-corrupted situation in Figure 12b, the reconstructed value fluctuates significantly. Compared with the measured value, the influence of noise on the reconstructed value is more obvious.

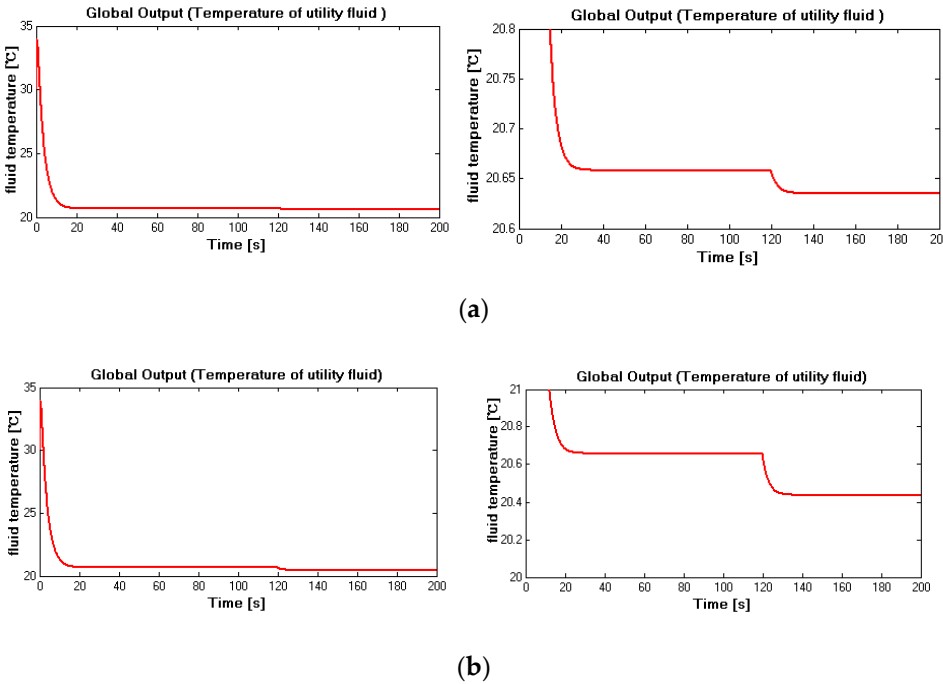

**Figure 10.** (**a**) Measured $T_H$ (noise-free situation) in case 2; (**b**) measured $T_H$ (noise-corrupted situation) in case 2.

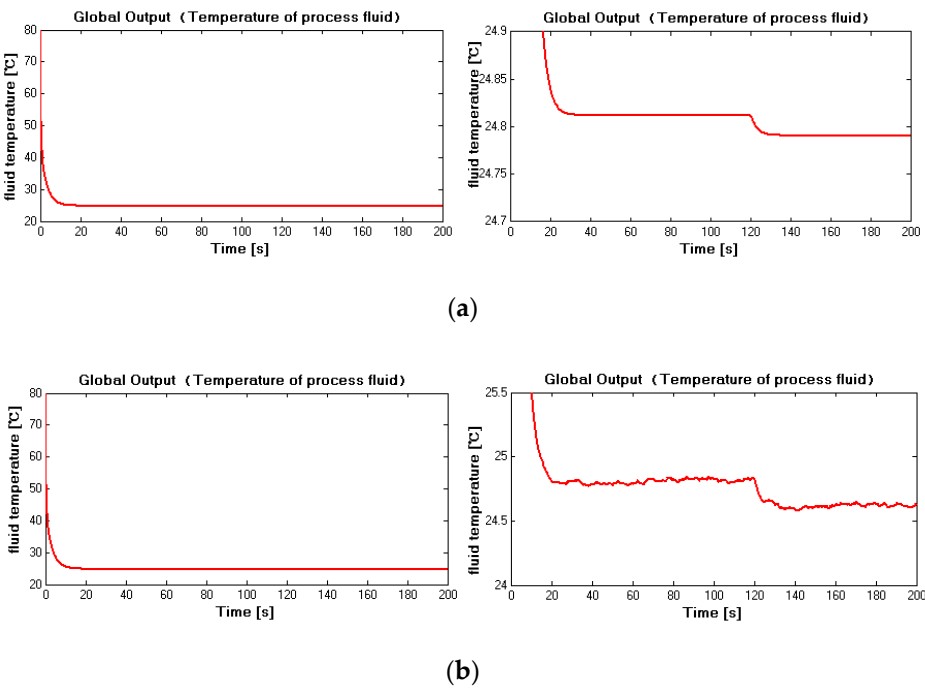

(**a**)

(**b**)

**Figure 11.** (**a**) Measured $T_C$ (noise-free situation) in case 2; (**b**) measured $T_C$ (noise-corrupted situation) in case 2.

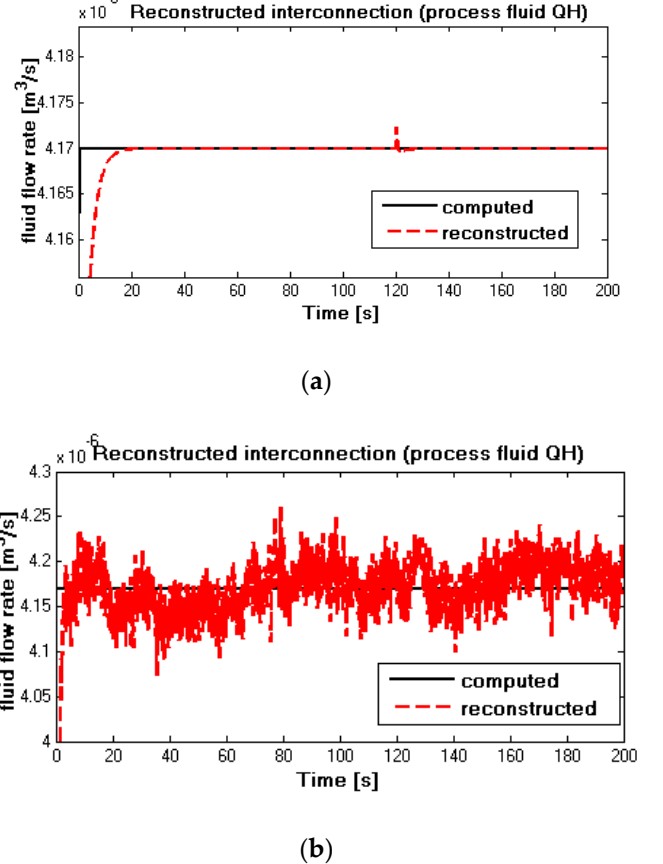

(**a**)

(**b**)

**Figure 12.** (**a**) Reconstructed and computed $Q_H$ (noise-free situation) in case 2; (**b**) reconstructed and computed $Q_H$ (noise-corrupted situation) in case 2.

For the utility fluid flow rate $Q_C$ in Figure 13a,b, it can be seen that the reconstructed values in red dash lines converge to the computed values in the black solid line after transient time. Because of changes at 120 s, reconstructed fluid flow rates vary accordingly and reach a new stable level. Similar to the process fluid, the reconstructed utility fluid $Q_C$ flunctuates significantly in the noise-corrupted situation in Figure 13b. Since the reconstructed values were retrieved from the global measured temperatures by higher-level subsystems, it can be seen that the variation of variables of the local subsystem has a significant impact on the global measurements, which is consistent with the hypothesis. Moreover, the simulation results show that the influence of noise on the reconstructed value is obviously stronger than the measured value.

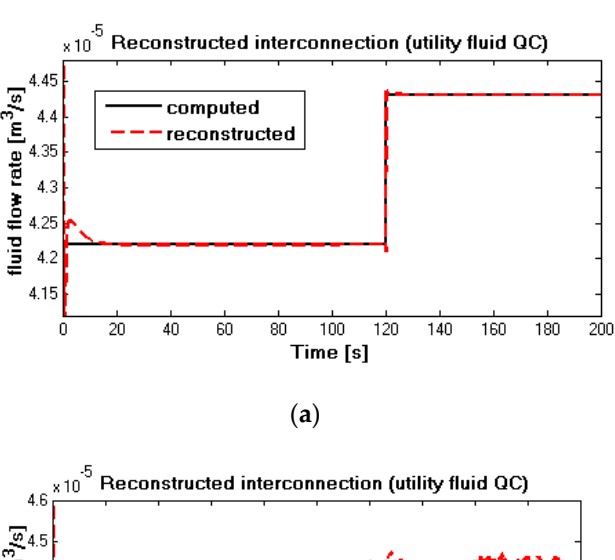

**(a)**

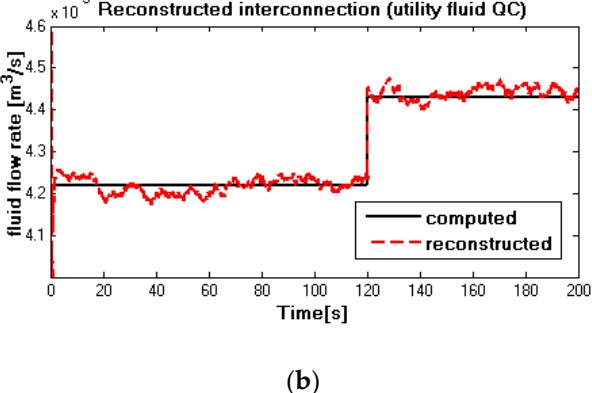

**(b)**

**Figure 13.** (**a**) Reconstructed and computed $Q_C$ (noise-free situation) in case 2; (**b**) reconstructed and computed $Q_C$ (noise-corrupted situation) in case 2.

Next, the output of the inverse of the high-level subsystem was fed back to the inverse of the low-level subsystem to reconstruct the original local input at the low-level subsystem, with the aim to identify the reconfigurability of the global interconnected system. The simulation results are shown in Figures 13–15.

As we can see from Figure 14a,b, recovered pneumatic pressure $p_{c1}$ in red dash lines follow the measured values in black solid lines after a short transient time. At time 120 s, the recovered value in the red dash line varies suddenly, but fortunately, it converges to the measured curve quickly. The reason for this variation is caused by the change of pneumatic pressure $p_{c2}$. In the noise-corrupted situation in Figure 14b, obvious impacts are observed.

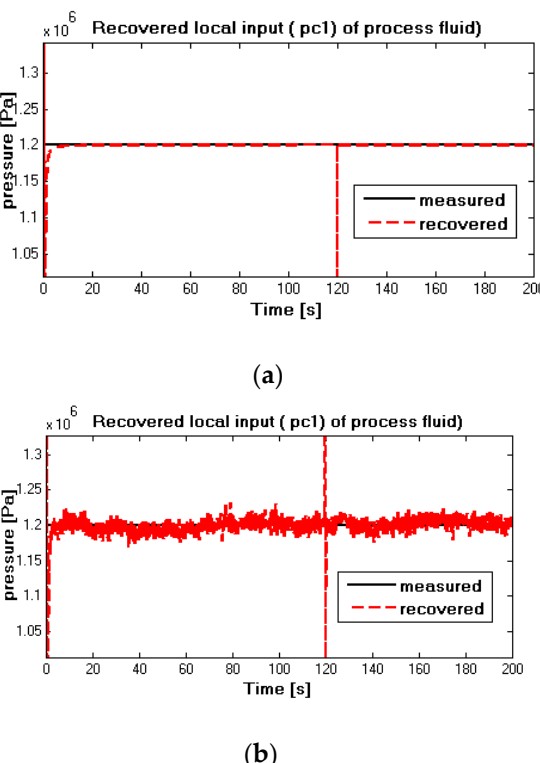

**Figure 14.** (**a**) Measured and recovered $p_{c1}$ (noise-free situation) in case 2; (**b**) measured and recovered $p_{c1}$ (noise-corrupted situation) in case 2.

The simulation results of pneumatic pressured $p_{c2}$ of utility fluid case are plotted in Figure 15a,b. Similar to the situation of $p_{c1}$ of the process fluid, under the noise-free case in Figure 15a, it can be seen that the recovered pneumatic pressured $p_{c2}$ in the red dash curve tracks the measured $p_{c2}$ in the black solid curve rapidly; at 120 s, the measured value increases suddenly, and after a relatively short transient time, the recovered value follows the measured one correctly again. From the simulation results, it can be concluded that the inverse interconnected system can uniquely recover the original local input of the interconnected system using global measurements. In other words, if the interconnected system is invertible, the local input has a significant and distinguishable impact on a higher level. For the noise-corrupted situation in Figure 15b, although the computation bias is relatively important, the recovered value in the red dash line can also track the measured value in the black solid line with acceptable accuracy.

It can be concluded that, if the invertibility of the interconnected system can be obtained, it is capable of recovering local inputs by the outputs at the global level with acceptable accuracy, which indicates that both its effectiveness and robustness are confirmed. Even though the noise power in the simulation is relatively small, and even if its influence on the measured value is very small, this small power noise will have a greater impact on the reconstruction value. In the experiment, it is found that the invertible cascade system cannot reconstruct the local input value well under the condition of a high-power noise.

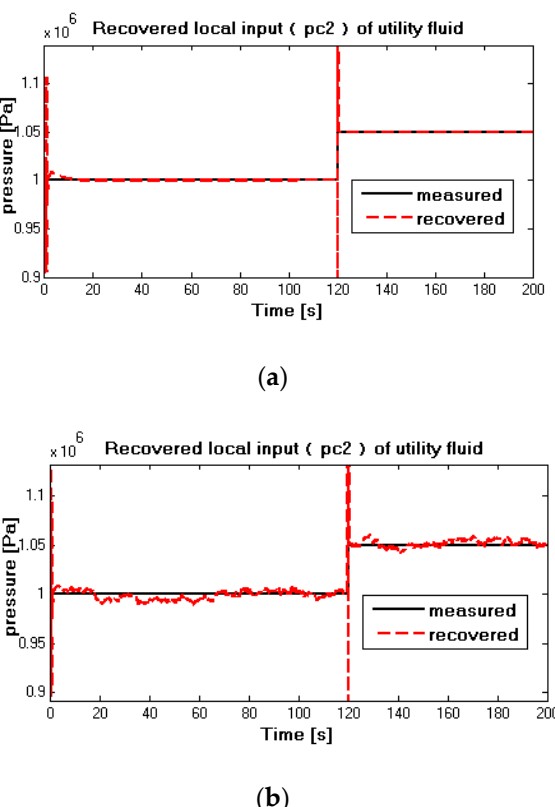

(**a**)

(**b**)

**Figure 15.** (**a**) Measured and recovered $p_{c2}$ (noise-free situation) in case 2; (**b**) measured and recovered $p_{c2}$ (noise-corrupted situation) in case 2.

## 6. Conclusions

In this paper, the invertibility of a nonlinear interconnected system consisting of two nonlinear affine subsystems was studied. A necessary and sufficient condition for guaranteeing the invertibility of the interconnected nonlinear system was established, involving the invertibility of individual subsystems. In order to recover the local input that yields the global output of the whole system, an algorithm was proposed that aimed at recovering the input uniquely in finite steps. Numerical simulations were included to confirm the effectiveness and robustness of the proposed methodology. Although there may be significant computation bias, especially when the output is corrupted by a high-power noise, the purpose is to confirm that the local input has a significant impact on the global level when the entire system is invertible. In this case, it allows the entire system to be monitored and analyzed on local subcomponents, but with global information.

However, in addition to the admirable features of the proposed methodology, there are open issues. An attractive direction is to establish a more constructive and relaxed condition for checking the invertibility of the interconnected systems. The goal is to verify the identifiability of the inputs (or unknown inputs), such as systems with more inputs than outputs, systems without a standard form, or with zero dynamic instability. The case where modeling uncertainties and measurement noise cannot be augmented into unknown input could be another interesting research direction in order to extend the applicability of the method proposed. Another problem to be solved is to verify the stability and sensitivity of the estimation error to prove that the required modeling information can be scaled without destroying the instability of the input reconstruction algorithm.

**Author Contributions:** M.Z. and Z.-T.L. conceived and designed the study. M.Z. carried out simulations and wrote the original draft. B.D. reviewed and edited the manuscript. All authors have read and agreed to the published version of the manuscript.

**Funding:** This research was funded by the National Natural Science Foundation of China, Grant No. 62003106 and No. 51867006, Talent Project of GZU (2018) 02. Key Lab construction project of Guizhou Province (2016) 5103.

**Institutional Review Board Statement:** Not applicable.

**Informed Consent Statement:** Not applicable.

**Conflicts of Interest:** The authors declare no conflict of interest.

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
