# Peer review of "On Invertibility of an Interconnected System Composed of Two Dynamic Subsystems"

_applsci, doi:10.3390/app11020596_

Round 1

Reviewer 1 Report

The paper analyzes the invertibility of interconnected systems. The goal is to examine whether the inputs at local subsystems can be recovered uniquely by output at global level and given initial states. It states necessary and sufficient conditions for guaranteeing invertibility of the interconnected system, which involves individual subsystems, is proposed. An algorithm is also proposed in order to recover local inputs that yield the global output of the entire system. Applying the algorithm the inputs can be recovered uniquely in finite steps. The paper presents several numerical examples.

The paper proposes extremely exciting and applicable results since in practice the modeling of complex physical systems often lead to a representation of interconnected systems. In my experience, however, models of physical systems always contain two components. One component is an uncertainty structure, which is caused by necessary simplifications. The other component comes from measurement noise or system noise. According to the reviewer, considering these components lead further research. I would also like to review the following research paper of the authors. However, it would be useful to present examples of what results can be expected in these cases.

Author Response

Dear Prof,

We feel great thanks for your professional review work on our paper. We acknowledge your comments and suggestions very much, which are valuable in improving the quality of our manuscript. As you are concerned, there are problems that need to be addressed. According to your nice suggestion, we have made revisions to our previous manuscript. We have added data to supplement our results and edited our paper extensively. We would like to know if there are still somewhere need to be amended.

Please see the attachment  to find the detail responses of your questions.

If you have any other questions about this paper, I would quite appreciate it if you could let me know at your earliest convenience, at [email protected].

Yours sincerely,

Mei ZHANG

Reviewer 2 Report

This is a good piece of work and its novelty lies in a new methodology to achieve the inevitability of an interconnected system. However, there are several aspects the manuscript can be improved.
1) The title and abstract should include the key contribution of the work. This is not a "proposed condition".
2) The references in the main text is a mess. For example, [4] appears before [2]; [12] before [10].
3) L528: Fig. 10 should be Fig. 9.
4) Figs. 12 & 13: part (a) is missing.
5) Lines 324 & 326: justified to right.
6) In the figures of the result section, the black line (computed) shall be brought to front to increase visibility.

Author Response

Dear Prof,

We feel great thanks for your professional review work on our paper. We acknowledge your comments and suggestions very much, which are valuable in improving the quality of our manuscript. As you are concerned, there are problems that need to be addressed. According to your nice suggestion, we have made revisions to our previous manuscript. We have added data to supplement our results and edited our paper extensively. We would like to know if there are still somewhere need to be amended.

Please see the attachment to find detail response.

If you have any other questions about this paper, I would quite appreciate it if you could let me know at your earliest convenience, at [email protected].

Yours sincerely,

Mei ZHANG

Reviewer 3 Report

In this paper, the invertibility of a nonlinear interconnected system consisting of two nonlinear affine subsystems is studied. A necessary and sufficient condition for guaranteeing invertibility of the interconnected nonlinear systems is proposed, which involves invertibility of individual subsystems. 

1) The article requires a massive editing of English language. I don't mean to be rude here, but given that this is an English journal, it is appropriate that the authors consult with an editor to better present their thoughts in a grammatically correct way.

2) Overall the problem description is very abstract. I would encourage the authors to describe the problem statement not just mathematically but also with physically realistic example for better readability. 

3) Given that this paper deals with nonlinear systems, it is always insightful to connect the work with linear systems theory. A obvious question which comes is if we linearize both the subsystems, under what conditions will that approach work out fine and what are the cases when that won't work and we will have to rely on the full nonlinear approach presented here. 

Author Response

(The authors gave the same response as above.)

Round 2

Reviewer 1 Report

Thanks for the detailed answers. The responses provided by the authors are satisfactory to me. The reviewer agrees that it is not necessary to cover all research directions in detail in this paper. However, it was important to mention these research directions.

Author Response

Dear Prof,

We feel great thanks for your professional review work on our paper.We are really grateful for your positive confirmation. According to your helpful suggestion, we have made a disccusion of different research focuses in the conclusion part.

If you have any other questions about this paper, I would quite appreciate it if you could let me know at your earliest convenience, at [email protected].

Yours sincerely,

Mei ZHANG

Reviewer 3 Report

I am satisfied with the significant improvement in readability of the article. One last suggestion would be to include the discussion in the author response (see Comment 3) on LTI systems in the manuscript itself. If not in the main article, it could be made a part of an Appendix section.  

Author Response

Dear Prof,

We feel great thanks for your professional review work on our paper.We are really grateful for your positive confirmation. According to your helpful suggestion, we have made a disccusion on LTI systems in the main article.

If you have any other questions about this paper, I would quite appreciate it if you could let me know at your earliest convenience, at [email protected].

Yours sincerely,

Mei ZHANG